# Dissecting the invasion of *Galleria mellonella* by *Yersinia enterocolitica* reveals metabolic adaptations and a role of a phage lysis cassette in insect killing

**Philipp-Albert Sänger, Stefanie Wagner, Elisabeth M. Liebler-Tenorio, Thilo M. Fuchs***

Friedrich-Loeffler-Institut, Institut für Molekulare Pathogenese, Jena, Germany

\* thilom.fuchs@fli.de

**Data Availability Statement:** All relevant data are within the manuscript and its Supporting Information files.

## Abstract

The human pathogen *Yersinia enterocolitica* strain W22703 is characterized by its toxicity towards invertebrates that requires the insecticidal toxin complex (Tc) proteins encoded by the pathogenicity island Tc-PAI$_{Ye}$. Molecular and pathophysiological details of insect larvae infection and killing by this pathogen, however, have not been dissected. Here, we applied oral infection of *Galleria mellonella* (Greater wax moth) larvae to study the colonisation, proliferation, tissue invasion, and killing activity of W22703. We demonstrated that this strain is strongly toxic towards the larvae, in which they proliferate by more than three orders of magnitude within six days post infection. Deletion mutants of the genes *tcaA* and *tccC* were atoxic for the insect. W22703 Δ*tccC*, in contrast to W22703 Δ*tcaA*, initially proliferated before being eliminated from the host, thus confirming TcaA as membrane-binding Tc subunit and TccC as cell toxin. Time course experiments revealed a Tc-dependent infection process starting with midgut colonisation that is followed by invasion of the hemolymph where the pathogen elicits morphological changes of hemocytes and strongly proliferates. The *in vivo* transcriptome of strain W22703 shows that the pathogen undergoes a drastic reprogramming of central cell functions and gains access to numerous carbohydrate and amino acid resources within the insect. Strikingly, a mutant lacking a phage-related holin/endolysin (HE) cassette, which is located within Tc-PAI$_{Ye}$, resembled the phenotypes of W22703 Δ*tcaA*, suggesting that this dual lysis cassette may be an example of a phage-related function that has been adapted for the release of a bacterial toxin.

## Author summary

Interactions of bacteria with invertebrates took place over a long period of time, and it is assumed that these animals are not only a reservoir for human pathogens, but have also shaped their evolution. We here report that *Y. enterocolitica* colonizes the midgut of *G. mellonella* via the insecticidal toxin complex Tc and subsequently migrates through the epithelial cell layer within the first 18 hours of infection. Once reaching the hemolymph, the pathogen grows to high cell densities to finally kill the insect. The massive proliferation

**Funding:** This study was funded by the Deutsche Forschungsgemeinschaft (DFG); FU 375/4-1,2 to TMF. PAS received a salary from the DFG. The funders had no role in study design, data collection and analysis, decision to publish, or preparation of the manuscript.

**Competing interests:** The authors have declared that no competing interests exist.

is fostered by a set of differentially regulated genes that constitutes an adaptation of *Y. enterocolitica* to the nutrient-rich environment encountered within the insect larvae. A successful infection not only depends on the insecticidal Tc, but also requires the activity of a phage-related lysis cassette that is involved in Tc release, probably without affecting bacterial cell integrity. We conclude that the investigation of oral invertebrate infections contributes to a better understanding of microbial pathogenicity.

## Introduction

Several bacteria are known to successfully colonize and infect invertebrates and to eventually profit from their bioconversion [1]. Key factors for insect infection are the insecticidal toxin complex (Tc) proteins, which were first purified from *Photorhabdus luminescens* [2]. Their oral insecticidal activity is comparable to that of the *Bacillus thuringiensis*- (Bt-) toxin [2]. Homologues of the Tc proteins have been described in insect-associated bacteria such as *Serratia entomophila* and *Xenorhabdus nematophilus*. 3-D structural analysis of the tripartite Tc suggests a 5:1:1 stoichiometry of the A, B and C subunits, with the A subunit forming a pentamer that associates with a tightly bound 1:1 sub-complex of B and C [3–5]. The TcA subunits are assumed to bind to the membranes of insect midgut cells and harbour a neuraminidase-like region that possibly confers host-specificity [5]. The B and C proteins of *P. luminescens* form a large hollow structure encapsulating the toxic and the highly variable carboxyl-terminus of TcC that has recently been demonstrated to ADP-ribosylate actin and Rho-GTPases [6–8]. The attachment of the Tc to the host cell membrane via glycans [9,10] is either followed by receptor-mediated endocytosis or release of the ADP-ribosyltransferase into the target cell [3,5,11]. In a pH-dependent manner, the TcA translocation channel is injected into the membrane of the host cell, and conformational changes subsequently allow the toxic component to be released into the translocation channel of TcA and from there into the cytosol [5,12].

Insecticidal Tc proteins are also present in the three human pathogenic *Yersinia* species (spp.) and in *Y. mollaretii* [13]. The pathogen *Y. enterocolitica* is characterized by a unique life-cycle, as some of its representatives are able to switch between two distinct pathogenicity phases that manifest in invertebrates or mammals [14]. Strain W22703 (biotype 2, serotype O:9) carries the highly conserved chromosomal pathogenicity island Tc-PAI$_{Ye}$ that encodes two regulators and TcA (*tcaA*, *tcaB1*, *tcaB2*), TcB (*tcaC*) and TcC (*tccC1*) toxin complex subunits. TcaA of *Y. enterocolitica* is essential for toxic activity towards larvae of the tobacco hornworm *Manduca sexta* and the nematode *C. elegans* upon oral uptake of cell lysates or living cells [15,16]. Tc mutants of strain T83 were shown to be attenuated in their ability to colonize the gut of orally infected mice [17], a finding that is in line with the broad cytocidal host spectrum of bacterial toxins [18]. *Y. enterocolitica* W22703 produces Tc proteins at environmental temperatures, but not at 37°C [15,16]. The thermodependent activation of insecticidal activity is mainly the result of an antagonism between the regulators TcaR2 and YmoA. The thermolabile TcaR2 is essential and sufficient to activate *tc* gene transcription at low temperatures [19], whereas the *Yersinia* modulator of virulence, YmoA, a Hha-like protein that interacts with the DNA-binding protein H-NS, represses *tc* gene transcription at 37°C [20].

Remarkably, two highly conserved phage-related genes, which are not clustered with other phage determinants, are present in all insecticidal pathogenicity islands identified so far in *Yersinia* strains [13]. These genes termed *holY* and *elyY* are located between *tcaC* and *tccC* (**S1A Fig**), and their products were recently shown to act as a holin and an endolysin (HE), respectively [21]. ElyY revealed an endopeptidase with high substrate specificity that cleaves yersinial

murein. Overexpression of HE lyses *Y. enterocolitica* at 37˚C, but not at lower temperature [21]. Upon deletion of the gene encoding the Lon A protease, which is involved in the thermo-dependent regulation of *Yersinia* virulence properties [22,23], we observed lysis of the *Y. enterocolitica* mutant at 15˚C, indicating that this enzyme controls the temperature-dependent activity of the HE cassette [24]. The biological role of this dual lysis cassette and its potential contribution to the insecticidal activity of *Y. enterocolitica* has not been elucidated.

Invertebrates are often used as alternative to mammalian models of infection to study bacterial or fungal pathogenicity and to evaluate therapeutic interventions [25]. Nematodes have successfully been used to identify virulence-related genes in a broad set of bacterial pathogens [16,26]. On the other hand, insect models such as *Galleria mellonella*, the Greater wax moth, are considered to provide further insights into pathogen-host-interactions due to their more elaborated innate immune system [27–29]. They also allow subcutaneous injection as well as oral application of bacteria and fungi, *in vivo* imaging of bacterial cells, monitoring of intracellular gene expression, detection of immune responses, and the investigation of antimicrobial drugs [13,30–34]. Subcutaneous infection of *G. mellonella* larvae demonstrated the insecticidal activity of several *Yersinia* spp. and identified the enterotoxin YacT of *Y. frederiksenii* [13,34,35]. Moreover, many *Yersinia* genes, including those contributing to virulence, are up-regulated at lower temperature [36–38], corroborating the hypothesis that invertebrates are a natural host of pathogens and may therefore have fostered their evolution [39].

Here, we established and applied oral force-feeding of *G. mellonella* larvae with *Y. enterocolitica* to study molecular details of the subsequent gastrointestinal and systemic infection. Survival assays, histopathology, immunofluorescence, determination of cell numbers in time courses, transcriptomics, and infections with mutants were performed to dissect the distinct stages of *G. mellonella* infection by the bacterial pathogen as well as the roles of *tc* gene products.

## Results

### Oral infection of *G. mellonella*

We first optimized the use of *G. mellonella* larvae (**Fig 1A**) for oral infection with *Y. enterocolitica*. Bacteria injected into the mouth of larvae are expected to pass along the digestive tract (**Fig 1B**) that they colonize to prevent excretion. Histological analysis of larvae of approximately two cm length revealed distinct sections of the digestive tract (**Fig 1C–1H**), namely mouth, esophagus, crop, intestine, rectum, and anus.

Force-feeding of bacterial cultures was carefully performed by injection with a Hamilton syringe. In the case of accidental tissue perforation, resulting in direct injection of bacteria into the hemocoel and thus early death of larvae, the larvae were excluded from the experiment. Owing to the larva weight of 150–200 mg, a maximum of 5 μl culture or medium were applied. Preliminary experiments to establish the optimal infection dose were conducted within a range of $10^2$ to $10^8$ colony forming units (CFU) of *Y. enterocolitica* and showed dose-dependent phenotypes. When $10^7$–$10^8$ CFU were injected, all larvae died between four and 14 h p.i., and the hemolymph of these cadavers contained *Y. enterocolitica* cells. No lethality and no melanisation, however, was observed after applying a lower dose of $10^2$ to $10^4$ cells at least until nine days p.i., and no *Y. enterocolitica* cells were detected in the hemolymph of the larvae. Finally, $10^5$–$10^6$ CFU revealed as optimal dose to perform infections of *G. mellonella* larvae with *Y. enterocolitica*, a value that corresponds well with those used in subcutaneous applications [31,40].

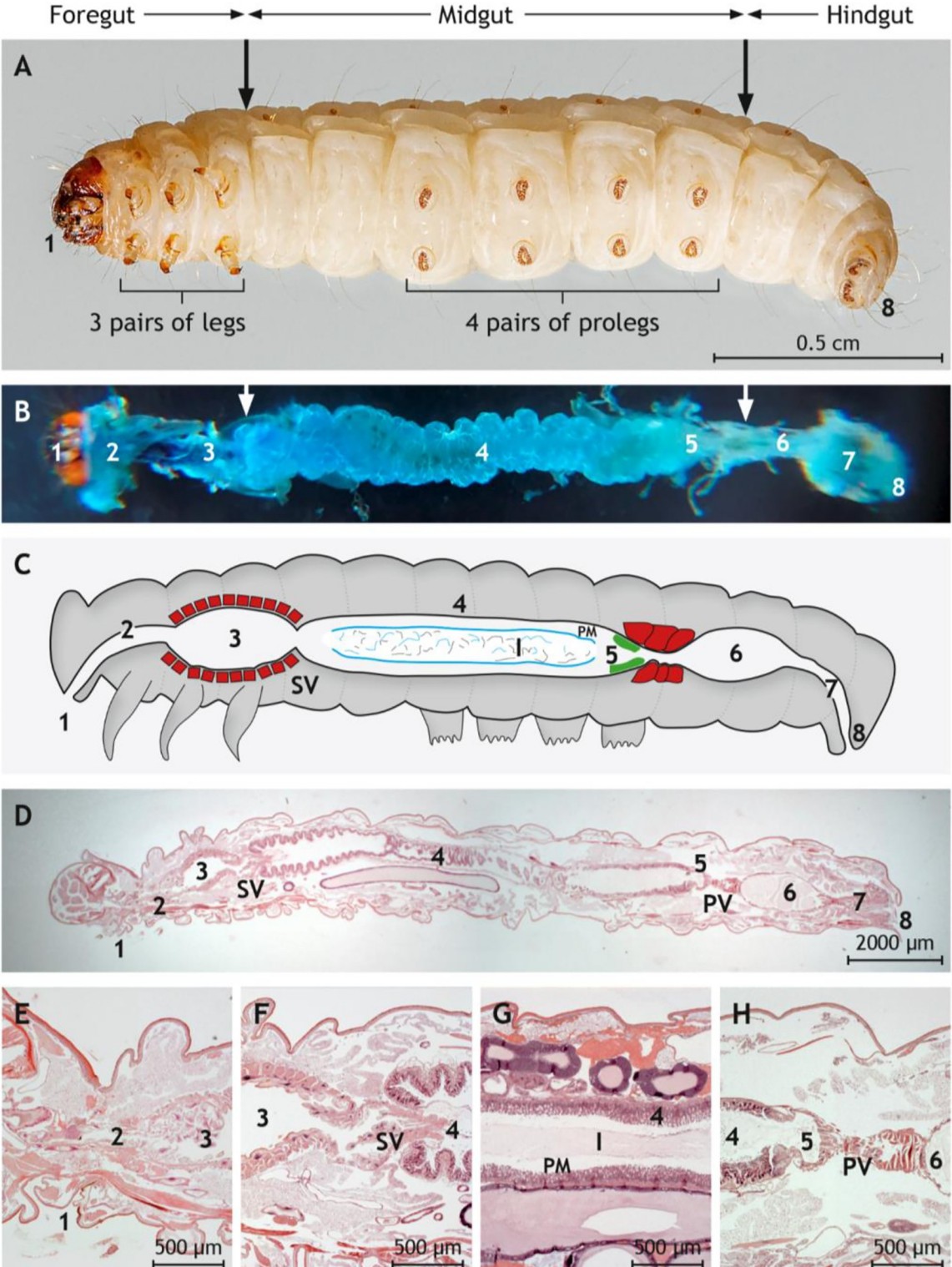

**Fig 1. Anatomy and histology of *G. mellonella* larvae with emphasis on the digestive tract.** (A) Underside of a *G. mellonella* larva. The foregut, the midgut, and the hindgut are indicated by arrows. (B) Dissected digestive tract after instillation with methylene blue. (C) Schematic drawing of the digestive tract. The stomadeal valve (SV) separates foregut lined by cuticular epithelium from midgut lined by glandular epithelium. The proctodeal valve (PV) is located between midgut and hindgut. The distinct epithelium cranial to the PV is labeled in green. The ingesta (I) in the midgut is covered by the peritrophic membrane (PM) and separated from the mucosa by the

ectoperitrophic space. The crop and both valves are surrounded by a thick layer of musculature (red). (D) Longitudinal and sagittal histological section along the middle through *G. mellonella*. (E) Magnification of mouth, esophagus, and crop lined by the cuticular epithelium and surrounded by muscle cells. (F) Magnification of the SV between crop and midgut. (G) Magnification of the midgut lined by glandular epithelium. The ingesta is surrounded by the peritrophic matrix (PM) and separated from mucosa by the ectoperitrophic space. (H) Magnification of the PV between midgut and hindgut lined by cuticular epithelium. Vacuolated columnar epithelial cells line the midgut cranial to the PV. (D)-(H) are paraffin sections stained by hematoxylin and eosin. Sections are indicated by numbers: 1 = mouth, 2 = esophagus, 3 = crop, 4 = glandular intestine, 5 = transition zone, 6 = cuticular intestine, 7 = rectum, 8 = anus. Photos of representative preparations are shown; the scales are indicated.

## TcaA and the lysis cassette are essential for the toxic activity of W22703 against *G. mellonella* larvae

Survival assays were performed with larvae of *G. mellonella* to further investigate the function of Tc-PAI$_{Ye}$ determinants in the interaction of *Y. enterocolitica* with insects. Recently, we demonstrated that subcutaneous infection of *G. mellonella* larvae with W22703 (LD$_{50}$ ~ $10^4$ cells) results in a killing rate similar to that of W22703 Δ*tcaA*, suggesting that the Tc plays a main role in the initial phases of infection rather than during systemic infection [13]. Here, we orally infected larvae with $5.7 \times 10^5$ CFU, $7.8 \times 10^5$ CFU, $5.6 \times 10^5$ CFU, and $6.2 \times 10^5$ CFU, respectively, of W22703 and its mutants W22703 Δ*tcaA*, W22703 ΔHE, and W22703 Δ*tccC*, and monitored the larvae for nine days. Larvae infected with *Y. enterocolitica* strain W22703 exhibited a significantly reduced survival rate with a time to death of 50% (TD$_{50}$) = 3.67 ± 1.12 days. In the infection experiments with the three mutants lacking *tcaA*, HE, and *tccC*, all larvae survived, corresponding to a challenge of the larvae with LB medium (**Fig 2**). To genetically validate that *tcaA*, HE, and *tccC* are essential for the toxicity of W22703 towards *G. mellonella*, we orally infected larvae with W22703 Δ*tcaA*/pACYC-*tcaA* ($9.0 \times 10^5$ CFU), $4.0 \times 10^5$ CFU (W22703 ΔHE/pACYC-HE), and W22703 Δ*tccC*/pBAD-*tccC* ($4.0 \times 10^5$ CFU). Due to the slight leakiness of the pBAD-promoter experienced recently [21], arabinose was not added to further induce *tccC* transcription. TD$_{50}$ of 2.91 ± 1.46 days, 1.83 ± 0.51 days, and 3.90 ± 0.41 days, respectively, were determined. Thus, the mutants harbouring recombinant plasmids that complement the deletion did not significantly differ in their insecticidal activity from that of the parental strain W22703 after one week, demonstrating that the *in trans* complementation of Δ*tcaA*, ΔHE, and Δ*tccC* fully restored the insecticidal phenotype of W2703. Taken together, these life span assays indicate that the two toxin subunits TcaA and TccC as well as the HE lysis cassette are strictly required for the oral toxicity of *Y. enterocolitica* strain W22703 towards the insect larvae.

In addition, we monitored the behaviour and the morphology of the larvae each day until six days post infection (p.i.), and again at day nine p.i. (**Fig 3**). Immediately after oral application, the control group infected with LB medium did not differ from the larvae infected with the three deletion mutants W22703 ΔHE, W22703 Δ*tcaA*, and W22703 Δ*tccC* with respect to motility and colour. In contrast, the application of W22703 and strains W22703 ΔHE/pACYC-HE, W22703 Δ*tcaA*/pACYC-*tcaA*, and W22703 Δ*tccC*/pBAD33-*tccC* resulted in a higher activity of the larvae. After 24 h, a strong melanization in the groups infected with W22703 and with the mutants harbouring complementing plasmids was observed. Similar to the untreated control groups, cocoons surrounded those larvae that had been infected with the deletion mutants, indicating that healthy individuals only are able to produce this protective housing. At days two to three, cocoon formation continued, whereas increasing numbers of larvae infected with W22703, W22703 ΔHE/pACYC-HE and W22703 Δ*tcaA*/pACYC-*tcaA* died. At days five to nine, morphological signs of infection did not further enhance. Nine days after infection, the first pupations events were observed.

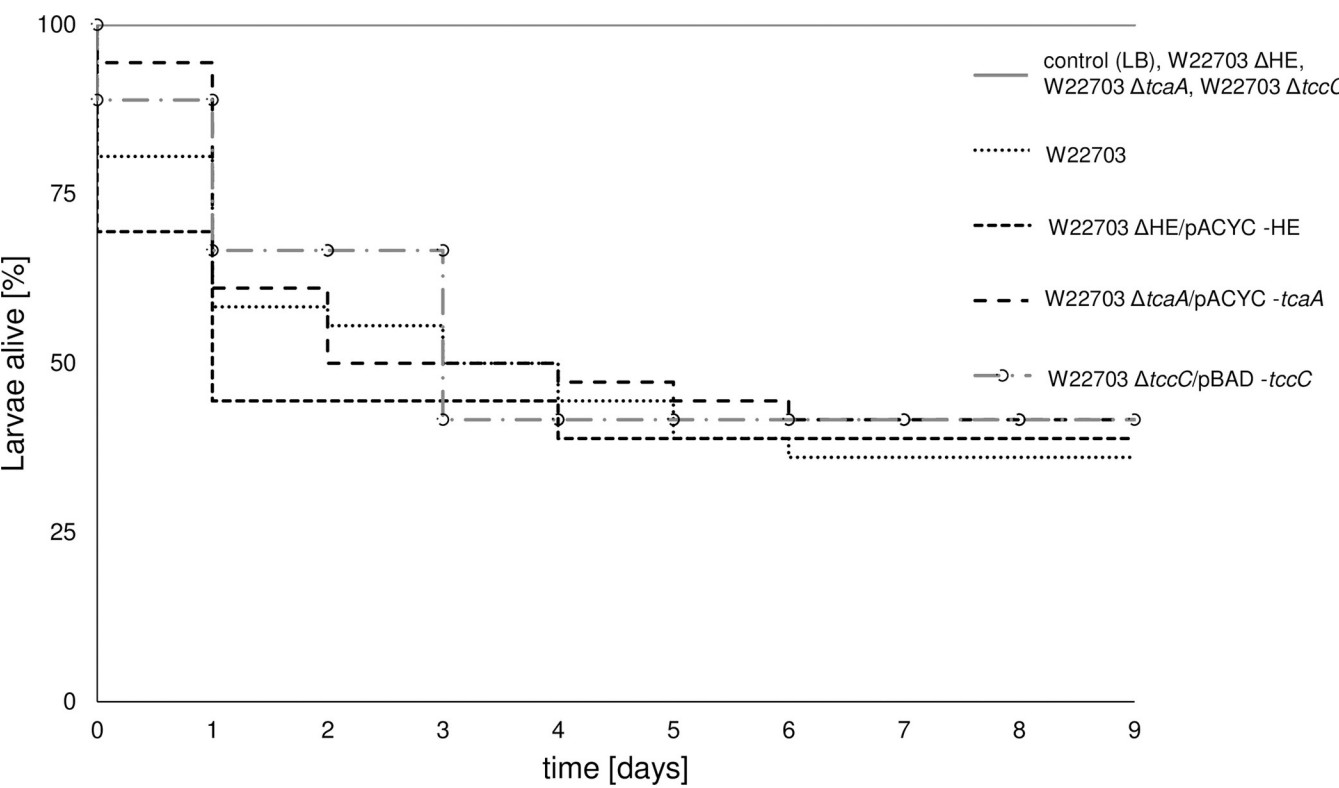

**Fig 2. Role of TcaA, HE, and TccC in insecticidal activity of W22703 towards *G. mellonella*.** Larvae were orally infected with W22703, its mutants lacking *tcaA*, HE, and *tccC*, and with mutants carrying the plasmids pACYC-HE, pACYC-*tcaA*, and pBAD-*tccC*. Application of LB medium served as control. Life span assays were performed for nine days, and the viability of the larvae was monitored each day to determine the survival rate of the larvae. The raw data were plotted by the Kaplan-Meier method. The Kaplan-Meier-plot is based on triplicates with 36 larvae in total per strain. The curves were compared to each other using the log-rank test, which generates a *p* value testing the null hypothesis that the survival curves are identical. Data were fit to exponential distribution. *p* values of 0.05 or less were considered significantly different from the null hypothesis (*p* value W22703 ΔHE/pACYC-HE = 0.0194; *p* value W22703 Δ*tcaA*/pACYC-*tcaA* = 0.0369; *p* value W22703 Δ*tccC*/pACYC-*tccC* = 0.0251). All graphs start at 100%.

## Orally injected W22703 mutants lacking TcaA, HE, or TccC are eliminated by *G. mellonella*

Proliferation within the insect host would indicate a successful infection by *Y. enterocolitica*. To determine the bacterial load over the time, we infected *G. mellonella* larvae with the same strains used in **Fig 2** and applied similar infection doses. One, three and six days p.i., the homogenate of six animals per time-point was plated on selective LB agar plates, and the CFU were enumerated. Strikingly, when we injected $9.0 \pm 0.2 \times 10^5$ CFU of mutant W22703 Δ*tcaA*, the total numbers of surviving bacteria rapidly decreased to $1.0 \times 10^3$ after one day and to eleven CFU after three days, and the strain was completely absent from the larvae six days p.i., probably due to passage through the gut followed by excretion (**Fig 4**). In contrast, the mutants W22703 ΔHE and W22703 Δ*tccC* that were applied with $4.0 \times 10^5$ CFU and $4.0 \times 10^5$ CFU, respectively, proliferated within the first day p.i. to $2.2 \times 10^6$ CFU and $2.8 \times 10^6$ CFU, but were not detected from day three on. This discrepancy suggests that TcaA is involved in adherence to epithelial cells and thus in midgut colonization, without requiring TccC. When larvae were infected with $4.0 \times 10^5$ CFU of the Δ*tcaA* and ΔHE mutants, and with $1.4 \times 10^6$ CFU of strain W22703 Δ*tccC*, all of which carrying recombinant plasmids that complemented the chromosomally deleted genes, the bacterial burden at days one to six p.i. increased approximately to that of the parental strain W22703 applied with $9.0 \times 10^5$ CFU, indicating a successful complementation of the gene deletions.

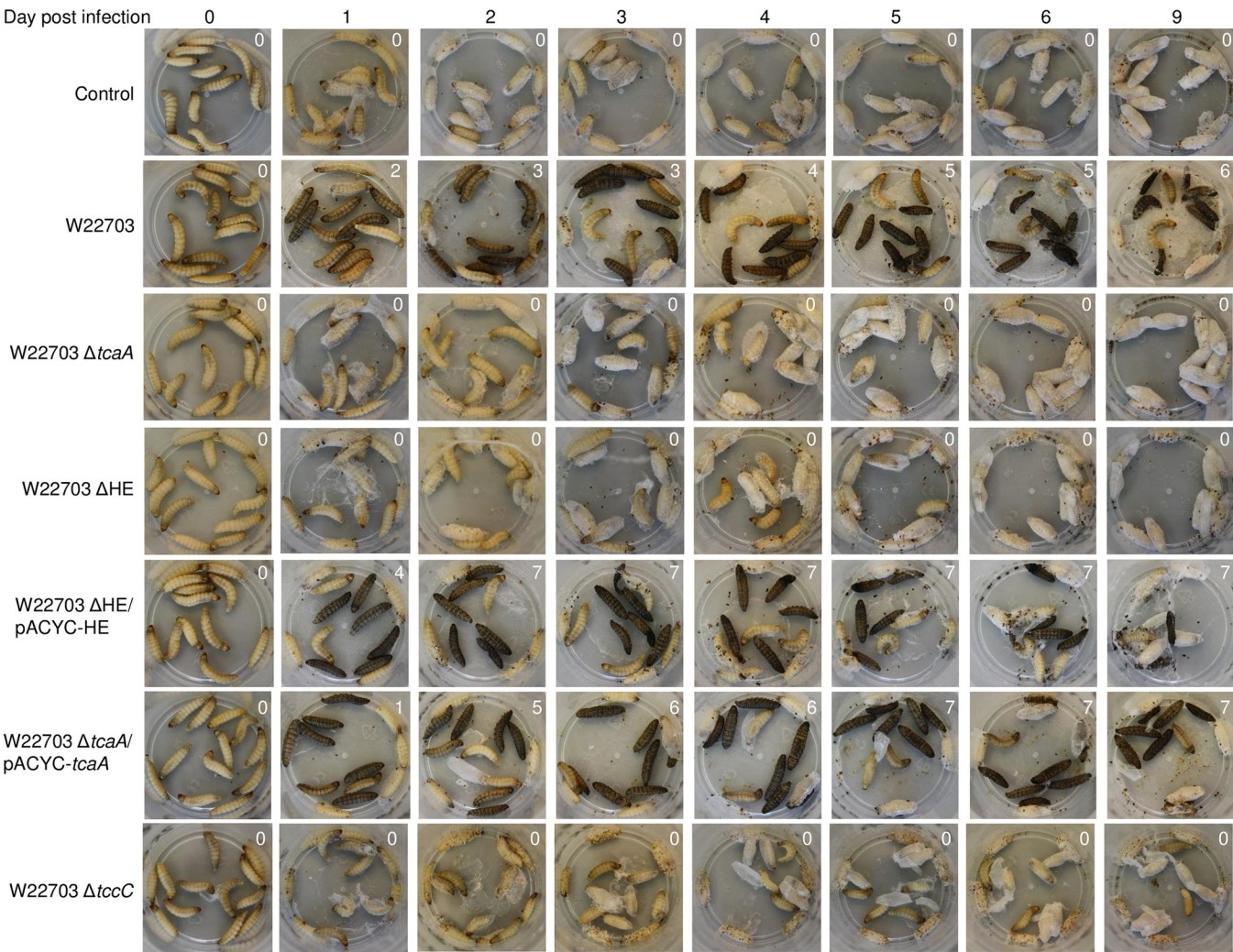

**Fig 3. External morphology of larvae following oral infection with W22703 and its mutants.** The photos illustrate the outcome of the experiment shown in **Fig 2**. Black animals were dead, anthracite ones still alive. The numbers in the upper right angle of each photo indicates dead animals at the respective time-point. Infections with W22703 Δ*tccC*/pBAD-*tccC* were not documented by photos.

These data show that only strains able to produce TcaA, the lysis cassette, and TccC possess the capacity to successfully colonize and to proliferate within the larvae.

## Time course of infection

The strongest proliferation of W22703 was observed between one and three days p.i. (**Fig 4**). We hypothesized that *Y. enterocolitica* starts its infection in the midgut and subsequently invades the tissues of *G. mellonella* larvae to proliferate in the hemolymph. To dissect the infection process preceding this multiplication in more detail, we performed a time course experiment using larvae infected with $6.3 \times 10^5$ *Y. enterocolitica* W22703 cells. Longitudinal sections through the middle of the larvae were prepared 4 h, 6 h, 12 h, 18 h, and 24 h p.i. and stained with FITC-conjugated *Yersinia*-antibodies. The histological analysis revealed that *Y. enterocolitica* is present in the midgut with increasing cell numbers until 12 h p.i. At this time-point, W22703 is mainly detected close to glandular epithelial cells of the midgut, implying a tropism of *Y. enterocolitica* for endodermal tissue (**Fig 5A**). Eighteen hours p.i., the gut appeared to be

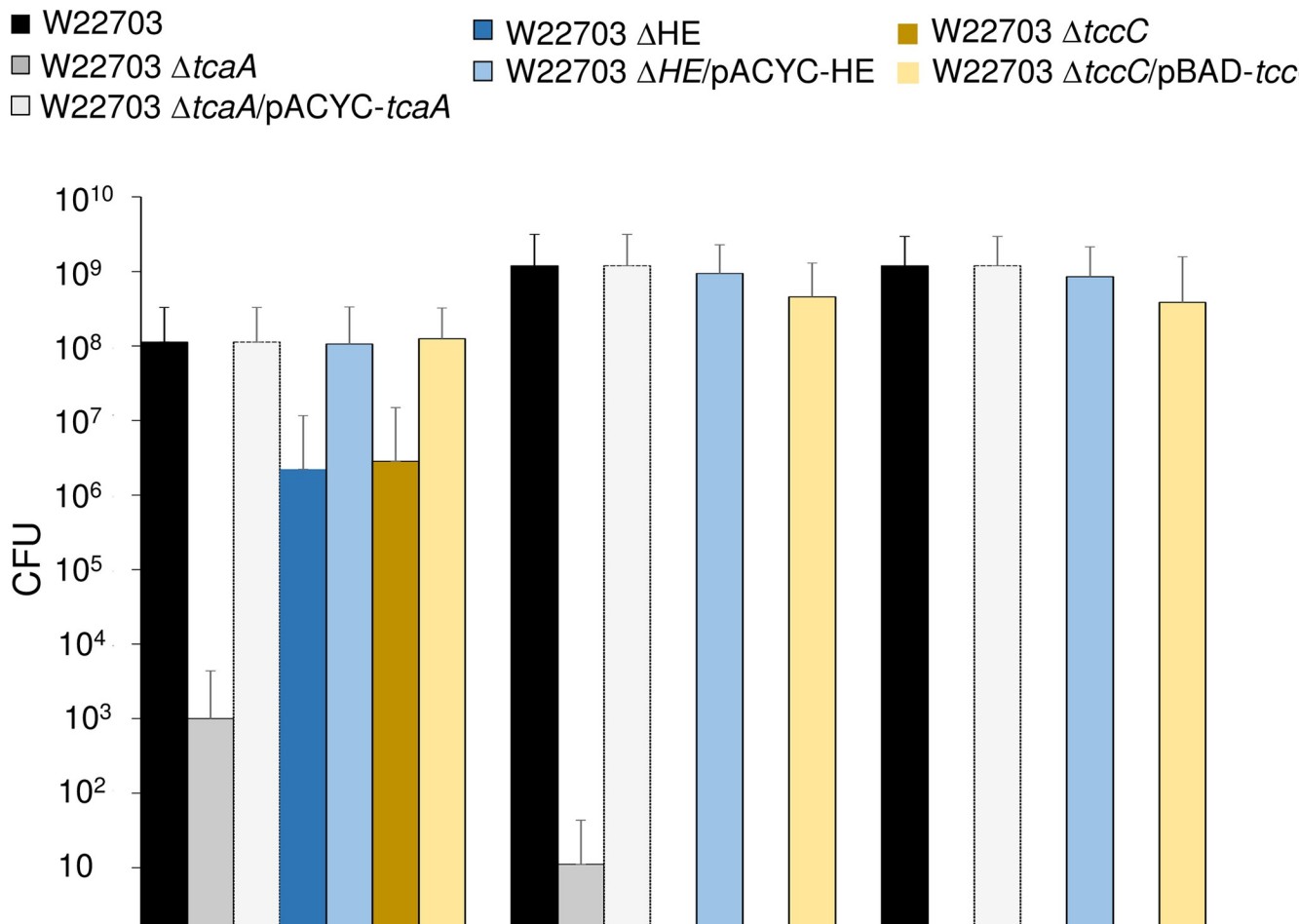

**Fig 4. Proliferation numbers of *Y. enterocolitica* W22703 strains in *G. mellonella* larvae.** Larvae were orally infected with W22703, W22703 Δ*tcaA*, W22703 Δ*tcaA*/pACAC-*tcaA*, W22703 ΔHE, W22703 ΔHE/pACYC-HE, W22703 Δ*tccC*, and W22703 Δ*tccC*/pBAD-*tccC*. Six animals per time-point were used, and the experiments were performed as triplicates, resulting in a total of 18 larvae per time-point per strain. Larvae fed with LB were used as a negative control. Standard deviations are indicated as error bars.

*Yersinia*-free, whereas a high number of cells is now detected in the hemolymph where W22703 proliferated with the next six hours to a high cell density. The larval tissues were completely overgrown with *Y. enterocolitica* 48 h p.i. when most animals were dead, indicating that the bacteria have started bioconversion of the cadaver (**Fig 5B**). To verify the finding that *Y. enterocolitica* W22703 cells are mainly found in the circulating fluid, we isolated 10 µl hemolymph from larvae infected with $1.6 \times 10^5$ bacteria. 24 h p.i., the animals showed clear signs of melanisation. The brownish hemolymph contained $2.3 \times 10^7$ to $4.7 \times 10^7$ CFU as detected on selective agar.

No fluorescence signal was obtained when we fed LB medium to larvae as a control (**Fig 5C**). Next, *E. coli* cultures were applied analogously to the experiments with *Y. enterocolitica*. Using an anti-*E. coli* antibody, *E. coli* cells, however, were not detected in the gut or in the hemolymph of *G. mellonella* 24 h after infection. To test the specificity of the antibody, *E. coli*

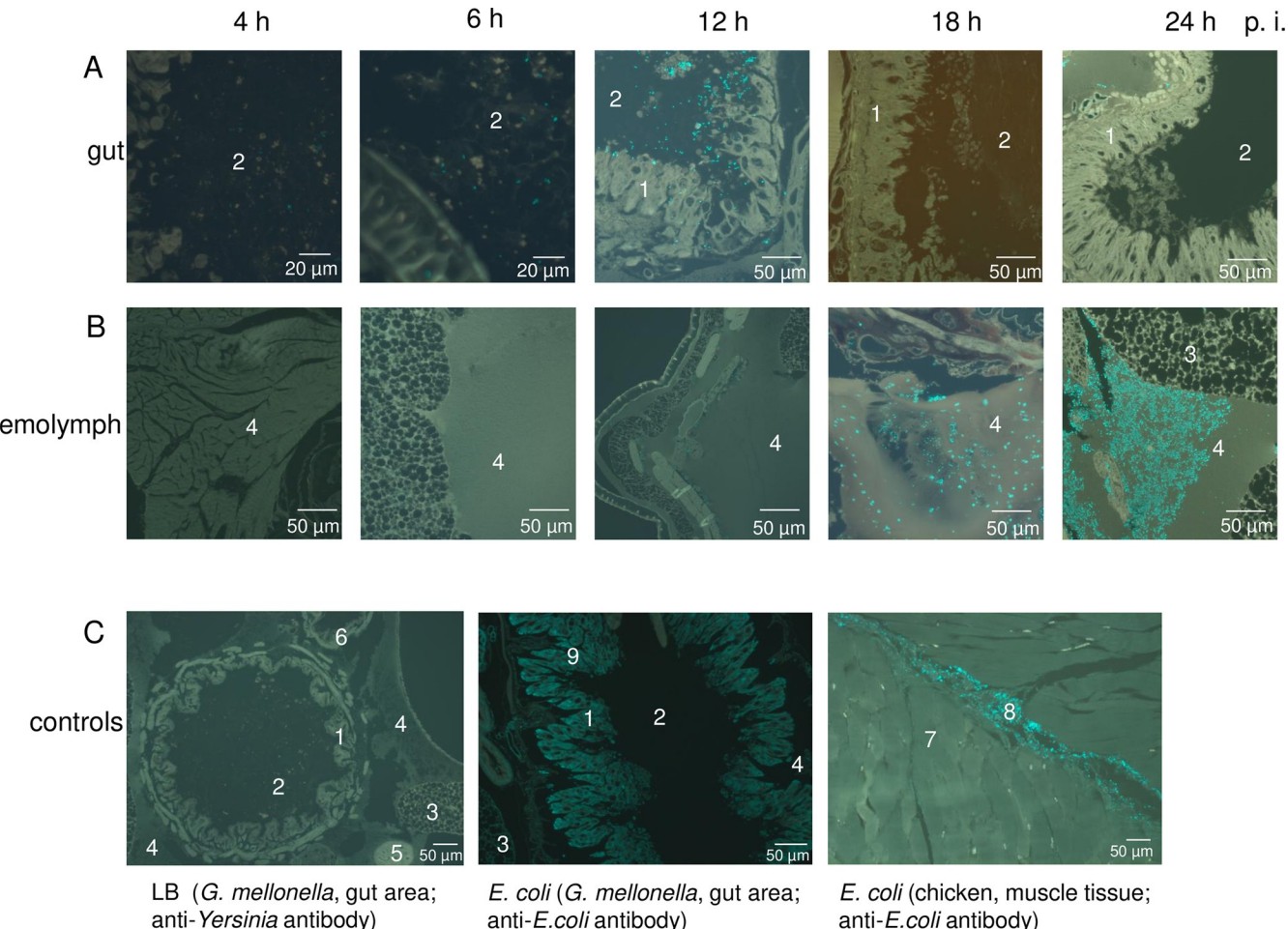

**Fig 5. Time course of *G. mellonella* infection by *Y. enterocolitica* W22703.** The tissue sections monitored by fluorescence microscopy show antibody-stained *Y. enterocolitica* cells in the (A) gut or (B) hemolymph of *G. mellonella* 4 h, 6 h, 12 h, 18 h, and 24 h after infection. (C) The controls depict the gut area of *G. mellonella* that were fed with LB (left) or infected with *E. coli* (middle) 24 h ago. The tissue sections were stained with a *Yersinia*-specific or an *E. coli*-specific antibody. Functionality of the anti-*E. coli* antibody was demonstrated by the application of *E. coli* into muscle tissue of chicken (right). Cyan-coloured areas in the gut area of *G. mellonella* are unspecific bonds of the anti-*E. coli* antibody. Representative preparations are shown; the scale is indicated. 1 = intestinal epithelium, 2 = intestinal lumen, 3 = fat tissue, 4 = hemolymph, 5 = appendix, 6 = Malpighian vessels, 7 = muscle cells, 8 = *E. coli*, 9 = antibody cross reactions.

DH5α cells were injected into muscle cells of a chicken leg and shown to be FITC-labeled 24 h p.i. The control experiments indicate that in contrast to *Y. enterocolitica*, *E. coli* is not able to survive in *G. mellonella*.

Taken these findings together, we demonstrated that *Y. enterocolitica* systemically infects *G. mellonella* larvae *via* colonization and penetration of the midgut epithelium to finally reach the hemolymph that allows massive proliferation of the pathogen.

## Strains lacking insecticidal genes do not enter the hemolymph

To determine whether or not the factors encoded on the insecticidal Tc-PAI$_{Ye}$ play a role during the infection process delineated above, *G. mellonella* larvae were orally infected with *Y. enterocolitica* W22703 ($6.3 \times 10^5$ CFU), W22703 $\Delta tcaA$ ($8.3 \times 10^5$ CFU), W22703 $\Delta tccC$ ($6.4 \times 10^5$ CFU), W22703 $\Delta$HE ($7 \times 10^5$ CFU), and W22703 $\Delta tcaR2$ ($7.8 \times 10^5$ CFU). In contrast to the parental strain W22703 that proliferated to high cell numbers in the hemolymph, no mutant cells were detected by FITC-staining of tissue sections made 24 h p.i. (**Fig 6A**).

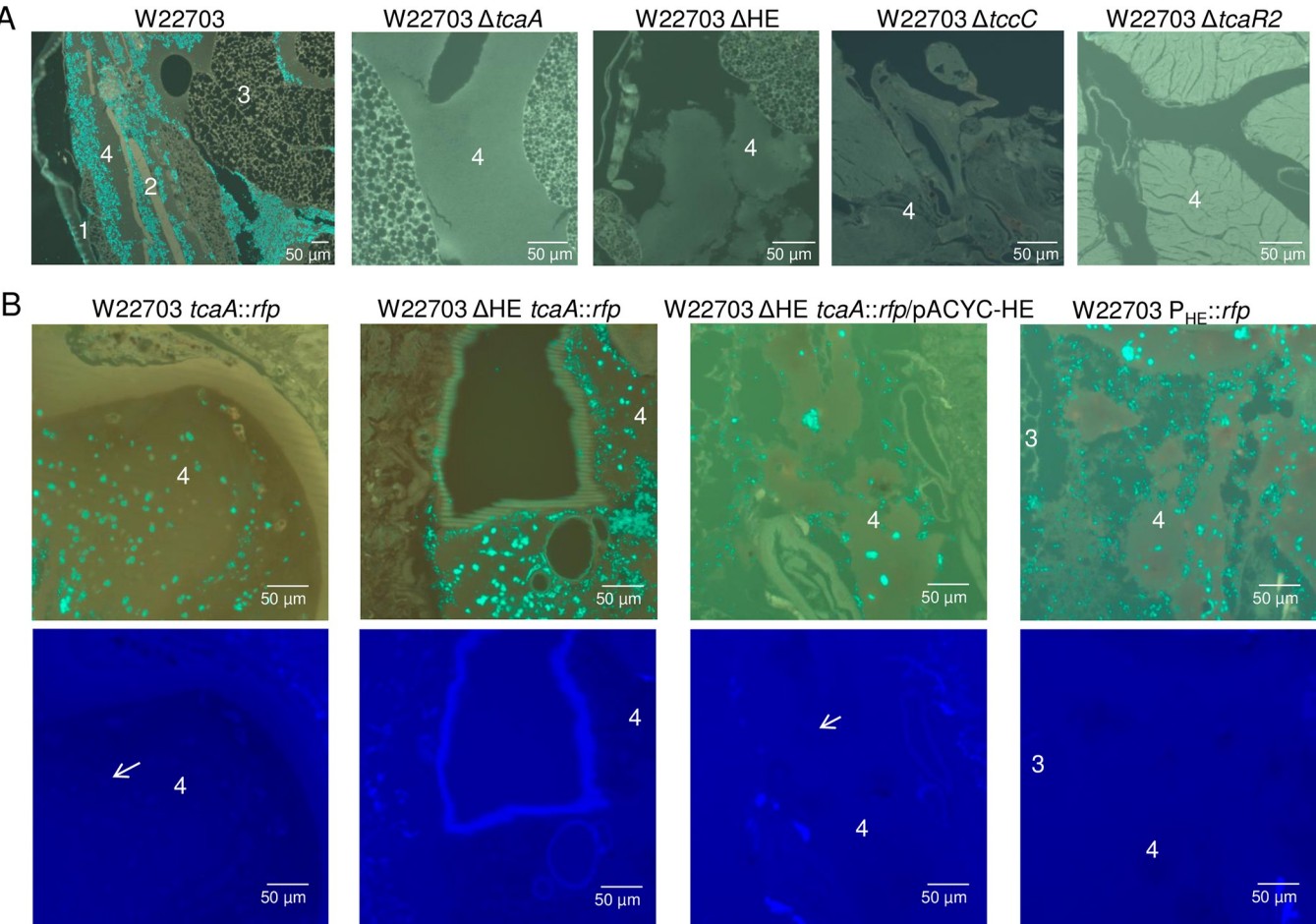

**Fig 6. Detection of W22703 and its derivatives in the hemolymph of *G. mellonella*.** All photos show the hemolymph area of *G. mellonella*. (A) *G. mellonella* tissue sections 24 h p.i. with W22703 and its mutants. Staining was conducted using the anti-*Yersinia* antibody. (B) Tissue sections of larvae 24 h p.i. with W22703 *tcaA::rfp*, W22703 ΔHE *tcaA::rfp*, W22703 P$_{HE}$::*rfp*, and W22703 ΔHE *tcaA::rfp*/pACYC HE. Staining was performed with the *Yersinia*-specific antibody (top row) or the anti-RFP antibody (bottom row), which was applied to the same tissue sections, to detect TcaA production as indicated exemplarily by arrows. Preparations of ten infected animals per strain were carried out. Photos of representative preparations are shown; the scales are indicated. 1 = exoskeleton, 2 = musculature, 3 = fat tissue, 4 = hemolymph.

These data confirm that the insecticidal Tc as well as the *tc* gene activator TcaR2 and the HE lysis cassette are required for full virulence of *Y. enterocolitica* W22703 towards *G. mellonella*. In particular, we hypothesize that the Tc-PAI$_{Ye}$ is responsible for midgut colonization and entering of the hemolymph.

## Detection of TcaA::RFP *in vivo* depends on a functional lysis cassette

To further investigate the role of the HE cassette in Tc release, we infected *G. mellonella* larvae with the reporter strain W22703 *tcaA::rfp*. Following tissue section and immunostaining with anti-*Yersinia* antibody and anti-RFP antibody, we detected TcaA in *Yersinia*-rich hemolymph areas of *G. mellonella* 24 h p.i. with W22703 *tcaA::rfp* and W22703 ΔHE *tcaA::rfp*/pACYC-HE (**Fig 6B**). In the absence of the lysis cassette, however, TcaA::Rfp was not detected despite the presence of W22703 ΔHE *tcaA::rfp* cells. To test whether or not the promoter of the lysis cassette is active *in vivo*, we infected *G. mellonella* larvae with strain W22703 P$_{HE}$::*rfp* that harbours a chromosomal transcriptional fusion of *rfp* with the HE promoter. Although this strain

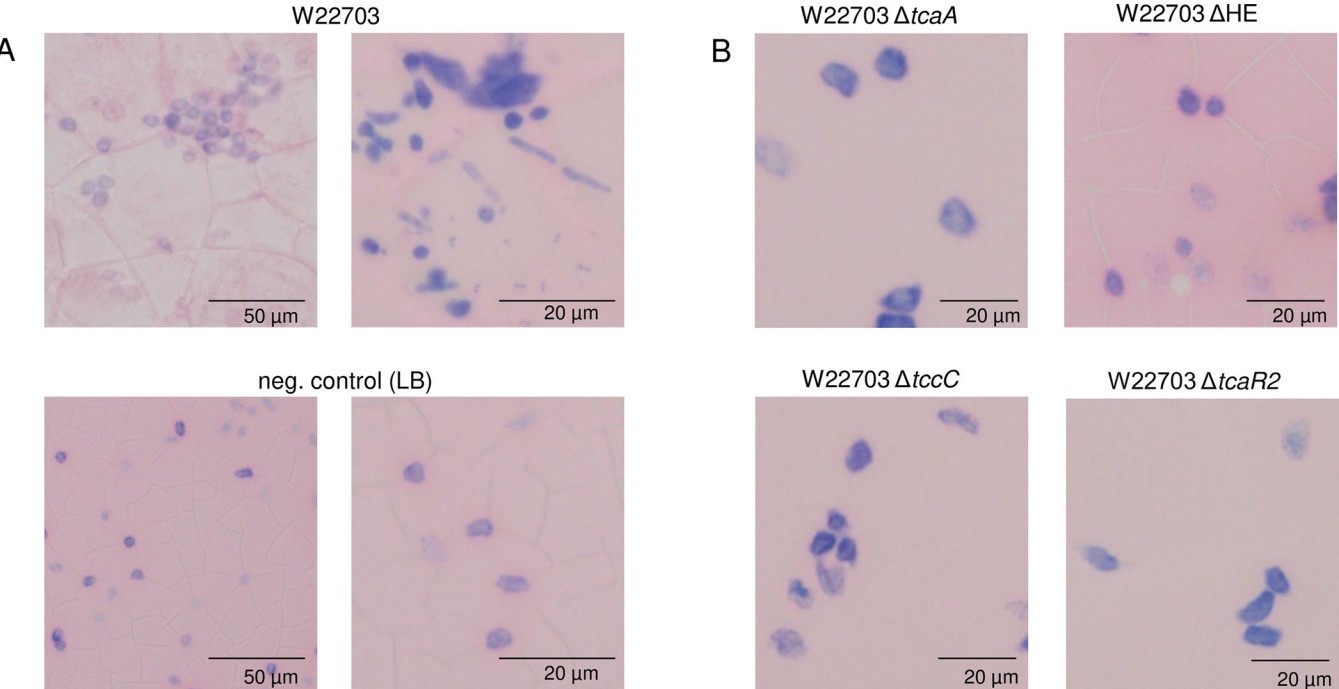

**Fig 7. Hemocyte morphology 24 h p.i.** (A) Hemolymph cell preparations from *G. mellonella* orally infected with $6.1 \times 10^5$ CFU of W22703 or treated with LB medium as control. (B) Hemocytes of larvae after application of four W22703 mutants, showing cell morphology similar to those of the controls. Hemocyte aggregation and deformation was visible only upon infection with W22703. Hemolymph preparations were fixed with methanol and stained by Giemsa solution. Photos of representative preparations are shown; cells vary in size, the scale is indicated. An Olympus BX53 microscope (Olympus Europa, Hamburg, Germany) was used.

densely proliferated within the hemolymph, we failed to stain RFP possibly due to no or weak $P_{HE}$ activity, an inference that is in line with the results of the transcriptome analysis (**S1B Fig**). Taken together, these data suggest that the HE cassette is responsible for the transport of the insecticidal Tc.

## Morphological effects of the TC on hemocytes

To investigate a possible activity of the insecticidal gene products on cells of the hemolymph, we monitored their morphology 24 p.i. For this purpose, hemolymph preparations of *G. mellonella* larvae were fixed with methanol and then stained by Giemsa solution. The hemocytes derived from W22703-treated larvae began to form aggregates in comparison with LB medium as control (**Fig 7A**). Cell agglutinations and a fading of the chromatin colour of the cell nucleus occurred more frequently. In contrast, hemolymph preparations of larvae one day after oral infection with W22703 mutants lacking *tcaA*, HE, *tccC*, or *tcaR2*, which encodes the activator of *tc* genes [19], showed hemocyte cell morphologies similar to those of the untreated controls (**Fig 7B**).

## *In vivo* transcriptome analysis

To delineate the transcriptional profile of *Y. enterocolitica* during infection of *G. mellonella*, we enriched and isolated *Y. enterocolitica* by immunomagnetic separation [41] from the larvae 12 h and 24 h after infection. These two time-points were chosen due to the results of the time course experiments (**Fig 5A and 5B**). Growth of W22703 cells in minimal medium with glucose as the carbon and energy source served as the reference condition. Following RNA

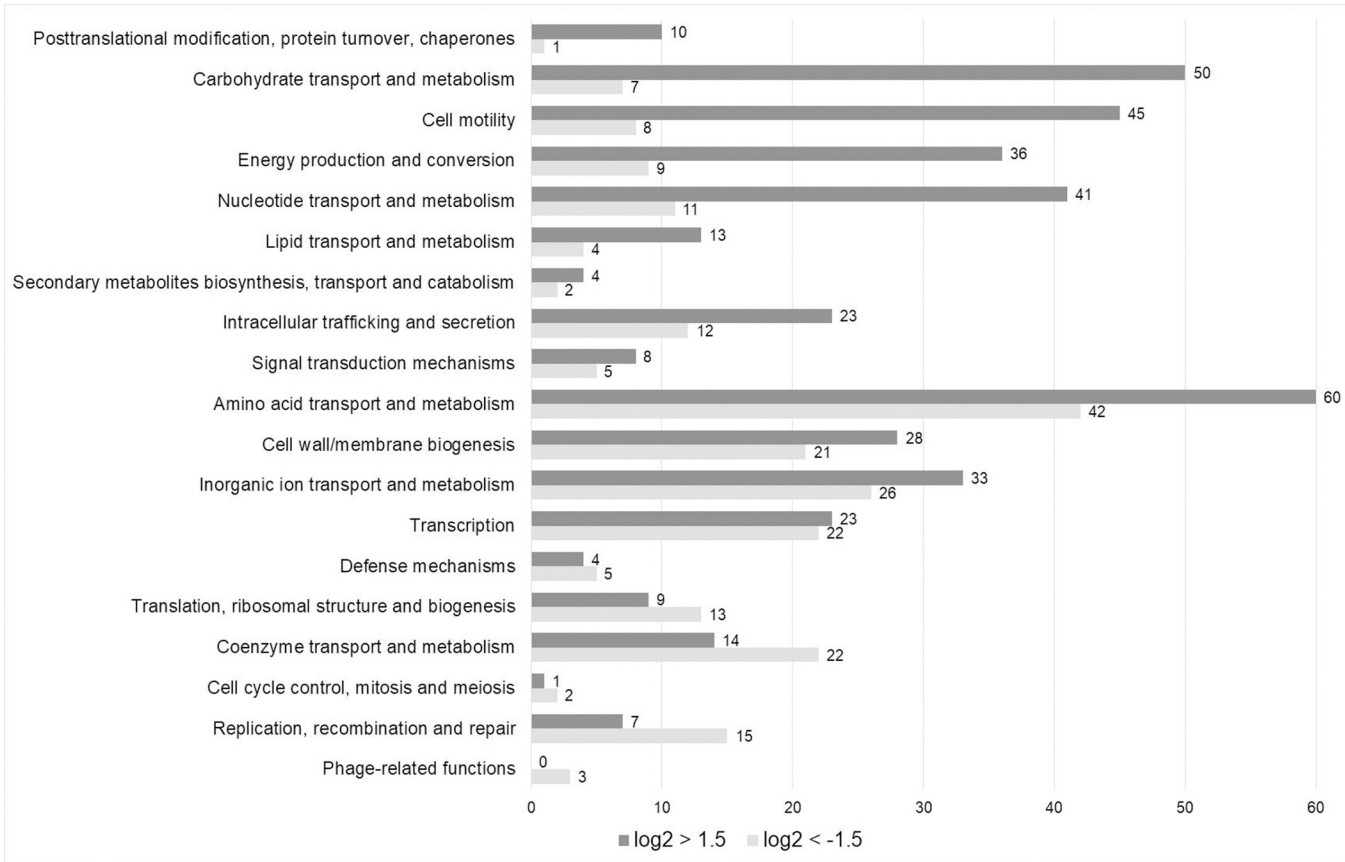

**Fig 8. COG categories of differentially expressed W22703 genes upon *G. mellonella* infection.** The numbers of down-regulated (light grey) and up-regulated (dark grey) genes in each COG category are indicated. The COG categories are ordered by descending ratio of up-to down-regulated genes. The data were taken from **S1 Table**.

sequencing and read analysis, we identified ~3,600 protein-encoding genes from a genome carrying ~4,000 genes [14], pointing to a 90% coverage of the transcriptional responses investigated here. Setting the threshold to a $\log_2$ FC $\geq$ |1.5|, we determined 524 non-redundant transcripts to be significantly more abundant and 301 to be less abundant in contrast to the control (**S1 Table**).

Owing to their important role in infection, colonization, and killing of *G. mellonella* demonstrated above, we first analysed the transcriptional activity of genes located on the insecticidal island Tc-PAI$_{Ye}$. Strikingly, the genes encoding the activator TcaR2 and the two Tc subunits TcaA and TcaB were strongly induced ($\log_2$ FC = 3.9, 5.5, and 3.2, respectively) 12 h p.i., but to lesser extent 24 h p.i. (**S1 Table**, **S1B Fig**). The endolysin gene *elyY* was significantly up-regulated after 24 h, but not after 12 h, implying a possible role of this enzyme in the release of the Tc. Beside the *tc* genes, the pathogen induced 15 virulence genes including those involved in iron acquisition, whereas a set of eight genes of this category was repressed, suggesting a role during infection of other host organisms including mammals.

Other main functional categories affected by a transcriptional switch upon larva infection were the metabolism and transport of amino acid and carbohydrates, resistance mechanisms, signaling, and motility (**Fig 8**). Within larvae, *Y. enterocolitica* down-regulated the biosynthesis pathways of methionine, isoleucine, leucine, histidine, tryptophan, and glutamate (**Table 1**). In contrast, determinants responsible for the transport of methionine, glycine, glutamate,

**Table 1. Selection of down (log₂ FC < -1.5, red)- and up (log₂ FC > -1.5, green)-regulated *Y. enterocolitica* genes 12 h and 24 h p.i.** AMP, antimicrobial peptide; P, phosphate; PTS, phosphotransferase system; BCAA, branched-chain amino acid; dep, dependent; n.s., not significant.

**virulence factors, iron metabolism**

| 12 h | 24 h | gene name | (predicted) function or product | COG |
|---|---|---|---|---|
| n.s. | n.s. | xaxB | α-xenorhabdolysin family binary toxin | R |
| | | hxuB | hemolysin secretion/activation | U |
| n.s. | | yqfA | hemolysin III | S |
| | n.s. | lktD | HlyD membrane-fusion protein of T1SS | M |
| | n.s. | mgtAC | magnesium transporting P-type ATPase | S |
| | n.s. | tcaR1 (aaeR) | toxin complex gene repressor | K |
| n.s. | | fepDG | ferric enterobactin transport | P |
| | n.s. | dinI | virulence protein MsgA | S |

**amino acid metabolism and transport**

| 12 h | 24 h | gene name | (predicted) function or product | COG |
|---|---|---|---|---|
| | | metABCFHIJLNQS | methionine biosynthesis/transport | E |
| n.s. | | mtnABCD | methionine biosynthesis/transport | E |
| n.s. | | map | methionine aminopeptidase | E |
| | | ilvABDEGMN | isoleucine biosynthesis | E |
| n.s. | | livFGM | BCAA transport | U |
| n.s. | | leuABCD | leucine biosynthesis | E |
| n.s. | | cysHIJM | cysteine biosynthesis from sulfate | EH |
| n.s. | /n.s. | cysCDGN | assimilatory sulfate reduction | F |
| n.s. | | cysAPTW | sulfate transport | P |
| n.s. | | hisDG | histidine biosynthesis | E |
| | /n.s. | trpEG/mtr | tryptophan biosynthesis/transport | EHU |
| n.s. | | gltBD | glutamate synthesis | E |
| /n.s. | | ddpABCDF; oppD | dipeptide transport | E |
| n.s. | | hutU | histidine degradation | E |

**cofactors**

| 12 h | 24 h | gene name | (predicted) function or product | COG |
|---|---|---|---|---|
| | | thiCEFGHIS | thiazole biosynthesis | H |
| n.s. | n.s. | bioABCDF | biotin synthesis | H |
| n.s. | n.s. | birA | biotin-operon repressor | HK |

**virulence factors, iron metabolism**

| 12 h | 24 h | gene name | (predicted) function or product | COG |
|---|---|---|---|---|
| | n.s. | srfB | virulence factor SrfB | S |
| | n.s. | cirA | iron-regulated outer virulence protein | P |
| n.s. | | shlB | hemolysin transporter/activator | U |
| | n.s. | hxuB | hemolysin secretion/activation | U |
| n.s. | | sodA | superoxide dismutase [Mn] | C |
| | /n.s. | tcaAB | insecticidal toxin complex subunits AB | K |
| | n.s. | tcaR2 | toxin complex gene activator | K |
| | | tpd | high-affinity $Fe^{2+}$ transport | P |
| | | ftnA | ferritin-1, iron-storage | P |
| | | sitABCD | chelated iron transport system | U |
| | | sitCD | chelated iron transport system | U |
| | n.s. | fepA | ferric enterobactin receptor | M |
| n.s. | n.s. | hemR | hemin receptor | P |
| | | foxA | ferrioxamine receptor | U |
| | | fbpA | $Fe^{3+}$-binding periplasmic protein | P |
| | | hasA | hemophore | C |
| | | phlA | hemolysin secretion/phospholipase | IU |

**amino acid metabolism and transport**

| 12 h | 24 h | gene name | (predicted) function or product | COG |
|---|---|---|---|---|
| n.s. | n.s. | metCIN | methionine metabolism and transport | E |
| n.s. | | metE | homocysteine methyltransferase | E |
| | n.s. | metY | o-acetylhomoserine (thiol)-lyase | E |
| | n.s. | putAPY | proline metabolism and transport | E |
| | | proX | glycine betaine-binding protein | E |
| n.s. | n.s. | gcvHPT | glycine degradation | E |
| n.s. | | ureCDEFG | urease subunit/accessory proteins | E |
| | | ureAB | urease subunits | E |
| n.s. | | cysK | cysteine synthase | E |

**cell wall, membrane, and envelop biogenesis**

| 12 h | 24 h | gene name | (predicted) function or product | COG |
|---|---|---|---|---|
| | | murPQ, amnK | anhydro-N-acetylmuramic acid utilization | G |
| | | ompD | outer membrane pore protein | M |
| | | galE | UDP-glucose 4-epimerase | M |
| | n.s. | ftsI | cross-linking of the peptidoglycan cell wall | M |
| | n.s. | rfaQ | lipopolysaccharide biosynthesis | M |
| | | agp | glucose-1-phosphatase | M |
| | | emtA | lytic murein transglycosylase | M |
| | n.s. | mltC | lysozyme inhibitor | M |
| | n.s. | | glycosyltransferase | M |
| | n.s. | lptD | oligogalacturonate-specific porin (KdgM) | M |
| n.s. | | slyB | outer membrane lipoprotein | M |
| n.s. | | mppA | periplasmic murein peptide-binding protein Pcp | E |

**signalling**

| 12 h | 24 h | gene name | (predicted) function or product | COG |
|---|---|---|---|---|
| | n.s. | cstA | carbon starvation protein A | T |
| n.s. | | dksA | DnaK suppressor protein | T |
| | n.s. | cheV | two component signalling adaptor domain | T |
| n.s. | | lsrABCDGFKR | autoinducer 2 import/regulation/degradation | G |
| n.s. | n.s. | traI | acyl-homoserine-lactone synthase | H |
| n.s. | | | LuxR family transcription regulatory protein | K |

**resistance**

| 12 h | 24 h | gene name | (predicted) function or product | COG |
|---|---|---|---|---|
| n.s. | | crcB | reduces fluoride concentration in the cell | U |
| n.s. | | tehB | tellurite resistance | PQ |
| | n.s. | yiiM | bleomycin resistance | S |
| | | | 6-N-hydroxylaminopurine resistance | S |
| n.s. | | cusA/czcA | resistance-nodulation-cell division | P |
| n.s. | | ohr | organic hydroperoxide resistance | O |
| | n.s. | bhsA | multiple stress resistance | S |

*(Continued)*

**Table 1.** (Continued)

| 12 h | 24 h | gene name | (predicted) function or product | COG |
|---|---|---|---|---|
| n.s. | | fthC | 5-formyltetrahydrofolate cyclo-ligase | H |
| n.s. | n.s. | lipB | octanoyltransferase (lipoate-dep. enzymes) | H |
| n.s. | n.s. | moaD | molybdopterin-convertion | H |
| **carbohydrate metabolism** | | | | |
| | | yadE | polysaccharide deacetylase | G |
| | | suhB | inositol-1-monophosphatase | G |
| | | ybbD | glycosyl hydrolase family | G |
| **cell wall, membrane, and envelop biogenesis** | | | | |
| n.s. | | | lipid A biosynthetic process | M |
| | | ftsI | peptidoglycan synthetase | M |
| n.s. | | murDF | peptidoglycan synthesis | M |
| | | mraY | peptidoglycan synthesis | M |
| | | ftsW | cell division | D |
| n.s. | | | O-antigen biosynthesis | M |
| | | ompC | outer membrane | M |
| n.s. | | lpxLM | lipid A biosynthesis | M |
| | | murJ | virulence factor mviN homolog | U |
| n.s. | | lnt | apolipoprotein N-acyltransferase | M |
| | | ttgA | membrane fusion protein | M |
| n.s. | | mltD | lytic murein transglycosylase D | M |
| n.s. | | tolA | cell envelope integrity | M |
| | | degP | protease | M |
| n.s. | | pgaBC | biofilm PGA synthesis | M |
| n.s. | | tolQR | outer membrane stability | U |
| **signalling** | | | | |
| n.s. | | | Lux regulon | K |
| n.s. | | | LuxR-family | KT |
| **resistance** | | | | |
| | | | threonine/serine transporter | U |
| 12 h | 24 h | gene name | (predicted) function or product | COG |
| | | tdcC | threonine/serine transporter | U |
| | n.s. | tdcB | threonine dehydratase | E |
| n.s. | | thrA | threonine synthesis | E |
| n.s. | n.s. | gltS | sodium/glutamate symport | P |
| | n.s. | gltI | glutamate/aspartate transport system | E |
| | | gltI | glutamate/aspartate-binding protein | ET |
| | /n.s. | glnPQ | glutamine transport | E |
| | | glnK | nitrogen regulatory protein P-II 2 | K |
| | | glsA | glutaminase 2 | E |
| n.s. | n.s. | aspA | aspartate ammonia-lyase | E |
| | n.s. | hisHIJMPQU | histidine transport/degradation | ET |
| | n.s. | sdaABC | serine dehydratase/transporter | E |
| n.s. | | glyA | serine hydroxymethyltransferase | E |
| | | trpDEG | anthranilate phosphoribosyltransf./synthase | F |
| | n.s. | tnaA | tryptophanase | E |
| n.s. | n.s. | argD | succinylornithine transaminase | H |
| n.s. | | acyG | subtilase | E |
| n.s. | | aroFP | aromatic amino acid synthesis/transport | E |
| n.s. | n.s. | speF | ornithine decarboxylase | E |
| n.s. | | prtA | secreted protease | E |
| n.s. | | pepD | dipeptidase | E |
| | n.s. | tesA | acyl-CoA thioesterase | E |
| | n.s. | ilvC | ketol-acid reductoisomerase | EH |
| n.s. | | livK | BCAA-binding protein | E |
| **cofactors** | | | | |
| | n.s. | tauABCD | taurine transport and metabolism | P |
| | | | | |
| n.s. | n.s. | kbl | 2-amino-3-ketobutyrate coenzyme A ligase | H |
| n.s. | | wrbA | flavoprotein | H |
| 12 h | 24 h | gene name | (predicted) function or product | COG |
| | n.s. | cutC | copper homeostasis | P |
| n.s. | | copA | copper-exporting P-type ATPase | P |
| | n.s. | yobA | copper resistance | S |
| n.s. | | ampC | β-lactamase | V |
| | n.s. | penP | β-lactamase | V |
| **carbohydrate metabolism and transport** | | | | |
| n.s. | | gpmA | phosphoglycerate mutase | G |
| | | glpF | glycerol uptake facilitator | U |
| | n.s. | glpDKQT | glycerol catabolism | G |
| | n.s. | sorBF | sorbose-specific PTS | G |
| | n.s. | agaCV | N-acetylgalactosamine permease | G |
| | n.s. | celC | lactose-cellobiose-chitobiose-specific PTS | G |
| | n.s. | ulaB | ascorbate-specific PTS subunit | G |
| | n.s. | cmtB | mannitol-specific cryptic PTS | G |
| | n.s. | uhpAC | regulation of hexose-P uptake | G |
| | /n.s. | xylFG | D-xylose import | G |
| | | rbsB | D-ribose-binding periplasmic protein | G |
| | | malEKMT | maltose operon (uptake and regulation) | S |
| | | lamB | maltoporin (maltodextrin permease) | E |
| | n.s. | iolABCDGET | inositol uptake and utilization | C |
| | n.s. | dctA | C4-dicarboxylate transport | U |
| | n.s. | dcuAB | anaerobic C4-dicarboxylate transporter | U |
| | n.s. | treBC | trehalose uptake and utilization | G |
| /n.s. | | srlAE | glucitol sorbitol-specific PTS | G |
| | n.s. | | sorbitol-6-P-2-dehydrogenase | IQ |
| | n.s. | mglABC | (methyl) galactoside/galactose transport | U |
| n.s. | | nagBE | N-acetylglucosamine-transport/utilization | G |
| | | fruB | multiphosphoryl transfer protein | G |

*(Continued)*

**Table 1.** (Continued)

| 12 h | 24 h | gene name | (predicted) function or product | COG |
|---|---|---|---|---|
|  |  | yoaE | heavy metal efflux pump | P |
|  |  | n.d. | tellurium-resistance | S |
|  | /n.s. | cusA/czcA | resistance-nodulation-cell division | P |
|  | n.s. | acrF | acriflavine resistance | V |
|  |  | mdlB | multidrug resistance | V |
|  | /n.s. | arnT/arnB | resistance to polymyxin and cationic AMP | I |
|  |  | zntA | cadmium, zinc and mercury-transport | P |
|  | n.s. | YPO2978 | chlorhexidine efflux transporter | S |
| n.s. |  | mdtK | multidrug efflux pump | P |
|  | n.s. | aaeB | p-hydroxybenzoic acid efflux pump | U |
|  | n.s. |  | metallo-β-lactamase | S |

| 12 h | 24 h | gene name | (predicted) function or product | COG |
|---|---|---|---|---|
|  | n.s. | yfiQ | CoA binding domain | H |
| **cell motility** | | | | |
|  |  | cheR | chemotaxis protein methyltransferase | H |
|  |  | cheD2 | methyl-accepting chemotaxis protein | NT |
|  | n.s. | tar | methyl-accepting chemotaxis protein | NT |
|  |  | cheABWYZ | chemotaxis | KT |
|  |  | motAB | chemotaxis | N |
|  |  | flaA | flagellin | N |
|  |  | flgBCDEFGIKL | flagellar hook and basal-body | N |
|  |  | fliJLST | flagellar proteins | N |
|  |  | fliDEFGHMN | flagellar assembly, hook, basal body, M-ring | N |
|  | n.s./ | flhCD | flagellar transcriptional activator | K |
|  |  | flhB | flagellar biosynthesis | N |
| n.s. |  | mqsR | motility quorum-sensing regulator mqsR | K |

| 12 h | 24 h | gene name | (predicted) function or product | COG |
|---|---|---|---|---|
|  | n.s. | pehX | exo-poly-α-D-galacturonosidase | G |
|  | n.s. | gmuD | 6-phospho-β-glucosidase gmuD | G |
| n.s. |  | fbaA | fructose-bisP aldolase class 2 | G |
|  | n.s. | fbp | fructose-1,6-bisP class 1 | G |
| n.s. |  | ptsG | glucose-specific PTS | G |
| n.s. |  | glgP | glycogen phosphorylase | G |
|  | n.s. | aglB | 6-phospho-α-glucosidase | G |
|  | n.s. | gmhA | phosphoheptose isomerase | G |
| n.s. |  | scrK | fructokinase | G |
| n.s. |  | gapA | glyceraldehyde-3-P dehydrogenase A | G |
| n.s. |  | fruK | 1-phosphofructokinase | H |
|  |  | glgC | glucose-1-P adenylyltransferase | H |
| n.s. |  | phnJ | synthesis of ADP-glucose | H |
| n.s. |  | scrY | sucrose porin | M |
|  |  | pckA | phosphoenolpyruvate carboxykinase | F |

glutamine, and histidine appeared at higher abundance *in vivo* in contrast to the control. This finding confirms the assumption that these amino acids are readily available in the surrounding hemolymph. The synthesis of thiamine was repressed, a finding that agrees with the reduced biosynthesis of isoleucine and leucine that require this cofactor. Another large number of up-regulated genes belongs to the categories of carbohydrate metabolism. *Y. enterocolitica* activated transporters including phosphotransferase systems and/or enzymes involved in the uptake and/or degradation of glycerol, sorbose, mannitol, ribose, xylose, inositol, trehalose, N-acetylgalactosamine, N-acetylglucosamine, sucrose, and glucitol/sorbitol. In addition, the pathogen induced a huge set of genes encoding factors of the nucleotide and lipid metabolisms as well as of the TCA-cycle, pointing to an increased metabolic activity and proliferation within insect larvae. Biofilm formation is not required for the virulence against insect as several biofilm and fimbriae producing genes were repressed and a biofilm repressor gene was transcriptionally activated. Motility appeared to play a pivotal role in insect infection by *Y. enterocolita* and by *Y. entomophaga* [29,42], because a large set of genes involved in flagella synthesis was up-regulated. Induction and repression of genes responsible for cell membrane biosynthesis pointed to major rearrangements in particular of the outer membrane within the larvae. Intriguely, several genes encoding factors involved in signaling were transcriptionally activated in the invertebrate, including the *lsr* operon responsible for autoinducer 2 import, regulation, and degradation, and *traI* encoding an acyl-homoserine-lactone synthase. No phage genes were induced, but 13 phage genes were repressed in larva-infecting *Y. enterocolitica* (**S1 Table**).

To summarize, the transcriptional activity of *Y. enterocolitica* in larvae of *G. mellonella* was mainly characterized by a drastic reprogramming of the energy, amino acid and carbohydrate metabolism, by an increase of motility and signaling molecules, and by cell membrane rearrangements.

## Discussion

To the best of our knowledge, the oral infection of *G. mellonella* by a bacterial pathogen performed here is of yet unprecedented resolution with respect to the molecular details, although force-feeding of *G. mellonella* was already established [43,44]. In contrast to subcutaneous injection in the use of insect larvae as model for bacterial virulence properties towards mammals, oral application mimics natural routes of infection that in particular take place during the bioconversion of animal cadavers by bacteria, fungi, and larvae [45]. In the present study, oral application of *Y. enterocolitica* demonstrated that the larvae are a powerful tool for dissecting the molecular steps required for a successful and lethal infection of *G. mellonella* by this enteropathogen. The distinct phases of infection identified here include survival in the gut, adhesion to and penetration of the midgut epithelial cell layer, massive proliferation within the hemolymph, hemocyte deformation, and insect killing. Despite an oral infection dose between $4.0 \times 10^5$ and $3.0 \times 10^6$ CFU, only few *Y. enterocolitica* cells were found in the gut according to cell counting and immunostaining, indicating that the majority of the bacterial cells is unable to maintain itself, and that W22703 does not substantially proliferate in the gut. A few *Y. enterocolitica* W22703 cells were seen in close proximity to glandular epithelial cells of the midgut. They probably cross the epithelial barrier via M-cells and migrate into the underlying tissues. The proliferation data shown in **Fig 4** allow the conclusion that TcaA enables the adherence to epithelial cells, whereas the enzymatic active subunit TccC plays a minor role here. The epithelial cell contact is compatible with the finding that the Tc of *Y. pseudotuberculosis* causes initial membrane ruffling of human colonic epithelial (Caco-2) cells [46]. The passage from the gut to the hemolymph occurs approximately 12 h to 18 h after ingestion, a delay reflecting the

time of the invasion process. Once they reach the hemocoel, the open circulatory system of the larvae, the pathogen exhibits a massive proliferation, pointing to excellent growth conditions and nutrient availability in this compartment [29]. Strain W22703 seems to withstand the phagocytic or growth suppressing activities of the hemocytes, probably as a result of Tc activity as suggested by changes of the hemocyte morphology in the presence of TcaA, TccC, and HE.

The *in vivo* transcriptome of *Y. enterocolitica* delineates its physiological and biochemical adaptations to the insect. The pattern of up- and down-regulated genes not only point to the relevance of distinct virulence factors, signalling, and motility for a successful infection, but also to the availability of numerous carbohydrates and amino acids in the insect body that fuels the metabolism and thus the proliferation of *Y. enterocolitica*. The carbon and energy sources are derived from either the diet or the host, such as glycerol, ribose, inositol, N-acetyl-galactosamine, or trehalose, which is present in the hemolymph as well as in honey, a component of *G. mellonella* feeding. The upregulation of genes involved in the uptake and degradation of sorbose and its reduced forms glucitol/sorbitol, of mannitol, of sucrose, and of xylose point to a specific metabolic adaptation of *Y. enterocolitica* to substrates fed by insects. Sorbose and mannitol are found in plant saps, and xylose is a monomer of hemicelluloses. Sucrose is one of the most concentrated nutrient available for sap-feeding insects [47]. In addition, the genes responsible for N-acetylglucosamine utilization suggests that chitin, a constituent part of the peritrophic matrices that line the inner surface of the gut in many insects [48], is not only degraded as a first step of colonization and invasion of the midgut epithelium, but metabolically utilized by the pathogen. When entering the *G. mellonella* larvae, strain W22703 in particular down-regulates the genes responsible for the synthesis of methionine, branched-chain amino acids, histidine, tryptophane, cysteine, and glutamate, indicating a sufficient availability of these amino acids within the insect. This is well in line with an increased capacity to import and degrade methionine, proline, glycine, urea, cysteine, threonine, glutamate/aspartate, serine, and histidine. Histidine is one of the most abundant free amino acids in the *Hyalophora gloveri* fat body [49]. It is worth to note that urease, inositol, and histidine degradation belong to metabolic properties that are common to *Y. enterocolitica* and *P. luminescens* [38]. The low temperature-dependent transcription of these and many other factors [37] fits to the proliferation of W22703 in insect larvae. Reprogramming of *Y. enterocolitica in vivo* activities also includes the increase of lipid import and degradation, and of energy production and conversion. The latter category reflects the massive proliferation in the hemolymph. In addition, the *in vivo* transcriptional pattern revealed the up-regulation of virulence factors mainly involved in hemolysis and iron scavenging, which are probably specifically directed against insects (**S1 Table**). A similar response was monitored in the interaction of *Y. entomophaga* with *G. mellonella* [35]. Autoinducer-2 import was also found to be up-regulated during insect infection. Given that more than 300 AI-2 regulated genes involved in regulation, metabolic activity, stress response and pathogenicity are known in *P. luminescens* [50], this points to an important role in signalling during insect infection.

We provide for the first time *in vivo* evidence for an involvement of the holin/endolysin cassette in Tc release and thus in nematocidal and insecticidal of *Y. enterocolitica* strain W22703. The dual lysis cassette is highly conserved in the genomes of *Yersinia* spp. where it is localised between genes encoding Tc subunits, and is also present in the Tc locus of *P. luminescens* [13]. These findings pointed to a functional role of the phage genes in the insecticidal activity of these bacteria, for example by cell lysis [51,52]. In *Y. pestis*, it was postulated that the release of the Tc is mediated by a type III secretion system (T3SS) [53], but this hypothesis was recently refuted [54]. This agrees with the lack of virulence plasmid pYV, which encodes a T3SS, in strain W22703 [14].

Thus far, there are only few examples of bacterial toxins that are released into the environment by phage-related factors. In *Serratia marcescens*, a holin and endopeptidase cassette were

identified to be required for the secretion of a chitinase [55]. A further example of a correlation between phage lytic genes and toxicity is the putative coupling of the λ phage lytic cycle and the release of the phage-encoded toxin Stx from Shiga toxin producing *E. coli* [56,57]. An N-acetyl-ß-D-muramidase similar to phage endolysins was shown to be essential for *Salmonella* typhoid toxin secretion [58]. The holin-like protein TcdE was shown to be required for *Clostridioides difficile* toxins TcdA and TcdB secretion via pore formation, and toxin release is independent of bacterial cell lysis [59,60]. The frequent neighbourhood of phage-related lysis factors to bacterial toxins and other secreted factors supports the hypothesis that protein release upon the activities of a dual lysis cassette evolved multiple times and defines a more widespread mechanism that was proposed to be termed the type 10 secretion system [61].

The holin/endolysin activity of *Y. enterocolitica* may allow the passage of the Tc through the inner membrane and the peptidoglycan layer. Given that no export signature such as the presence of a signal peptide for the sec-dependent pathway was found in the toxin subunit sequences of *Y. enterocolitica*, it may be assumed that the holin acts as a pore for both the endolysin and the Tc proteins. The Tc proteins then possibly exerted their toxic activity via release into the extracellular space or by being bound on the surface of the bacterial cells, as demonstrated for both YitA and YipA of *Y. pestis* [62], followed by the internalisation of the whole Tc or subunit C into the host cell [5]. Alternatively, the Tc may be released upon cell lysis triggered by the HE cassette. However, the fact that *Y. enterocolitica* cells are less susceptible to HE-mediated cell lysis at 15°C in comparison with 37°C using an artificial overexpression plasmid system does not support this hypothesis [24]. Moreover, the lysis cassette is very tightly repressed *in vitro* [21], a finding that was confirmed by the *in vivo* transcriptome analysis in the present study that revealed slight inductions of *elyY* ($\log_2$ FC = 1.02) and *hlyY* ($\log_2$ FC = 0.87)(**Fig 1B**). However, these data do not exclude the possibility that a partial subpopulation of *Y. enterocolitica* undergoes cell suicide and releases the Tc for the benefit of cells not expressing the HE cassette. Such a liberation of cell contents via endolysin-mediated cell lysis has been proposed to explain membrane vesicle formation and biofilm development of *Pseudomonas aeruginosa* [63]. A detailed model of the molecular mechanism underlying the Tc release from *Y. enterocolitica*, however, remains to be elucidated.

## Materials and methods

### Bacterial strains, plasmids and growth conditions

Bacterial strains and plasmids used in this study are listed in **S2 Table**. All cultures were grown in Luria-Bertani (LB) broth (10 g/l tryptone, 5 g/l yeast extract, 5 g/l NaCl), in minimal medium consisting of M9 medium supplemented with 2 mM MgSO$_4$, 0.1 mM CaCl$_2$ and 55.5 mM (1% w/v) glucose or on LB agar (LB broth supplemented with 1.5% agar). *E. coli* was grown at 37°C, and *Y. enterocolitica* at 30°C or as indicated. For growth studies, overnight cultures were diluted 1:1,000 in 200 ml flasks with 50 ml medium and incubated under vigorous shaking until cells reached stationary phase, or in microtitre plates with 200 μl medium per well. If appropriate, kanamycin (50 μg ml$^{-1}$), streptomycin (50 μg ml$^{-1}$), chloramphenicol (20 μg ml$^{-1}$), tetracycline (12 μg ml$^{-1}$), nalidixic acid (20 μg ml$^{-1}$), or arabinose as indicated were added to the media. Each plate pack was tested for antibiotic activity using sensitive bacterial strains.

### General molecular techniques

DNA manipulation and isolation of chromosomal DNA was performed according to standard procedures [64], or to the manufacturer's protocol. Polymerase chain reactions (PCR) were carried out as described recently [19]. Chromosomal DNA (100 ng) was used as template for PCR amplification.

## Construction of mutants and plasmids

The construction of non-polar deletion mutants is exemplified here for the lysis cassette. Two fragments of 984 bp and 751 bp were amplified with the oligonucleotide pairs Hol.delF1/Hol.delR1 and Endo.delF2/Endo.delR2, and ligated *via* the introduced *Eco*RI sites. Following nested PCR with the oligonucleotides Hol.nestedAB and Endo.nestedCD and the ligation mixture as a template, the resulting fragment was cloned into pKNG101 *via Bam*HI, giving rise to pKNG101ΔHE. The resulting construct was transformed from SM10 into W22703 by conjugation, and a selection procedure was performed as described recently [15]. Streptomycin-sensitive clones were screened by PCR to identify mutant W22703 ΔHE. The deletion was confirmed by sequencing. To complement W22703 ΔHE, the complete coding sequence of the *holY/elyY* cassette and 250 nucleotides of its upstream sequence were amplified and cloned into pACYC184 *via Eco*RI in DH5α. In the resulting plasmid pACYC184-HE, the direction of *holY/elyY* transcription corresponds to that of the disrupted plasmid gene encoding the chloramphenicol acetyltransferase. The gene *tccC* was cloned into pBAD33 *via Sac*I and *Pst*I. All oligonucleotides used here are listed in **S3 Table**.

## Bioassays

Larvae of *G. mellonella* were obtained from the Klee-Gartencenter + Zoo (Jena, Germany), and stored for less than one week at 15˚C. Bacterial strains were grown to early stationary phase (optical density at 600 nm [$OD_{600}$] ~ 1.35) at 20˚C (*Yersinia* spp.), or at 37˚C (DH5α). Larvae of instar 5–6 [65], 2–3 cm length and 120–150 mg in weight were used. Before application, the larvae were placed on ice for about 10 min as a light anesthesia. Five µl of the respective bacterial culture or its dilution in LB medium were applied orally using a Hamilton syringe (Hamilton 702 RN, 25 µl) with a very small and blunt cannula (needle gauge 33). For the application, the cannula was inserted over a maximal length of 1 mm into the space between the labrum and labium, and the liquid was applied slowly until being completely absorbed by the insect. During the infection experiments performed at 20˚C-22˚C, animals were not fed to ensure similar conditions.

Infection doses were determined by plating serial dilutions of the suspensions used for oral application. Selective agar plates with LB or *Yersinia* selective medium (Schiemann CIN medium, Oxoid, Wesel, Germany) were incubated at 30˚C for 24 h. Infected larvae were kept up to nine days in the dark at the temperature indicated, and the numbers of killed and alive larvae were enumerated each day. Larvae were considered dead if they failed to respond to touch. The $TD_{50}$ was calculated using the dose-response curve (drc) package of the R software. To recover bacteria from the larvae, the larvae were surface sterilized with 70% ethanol, washed in $H_2O$ and homogenized in a mortar. The homogenous mass was suspended into 1 ml LB, rigorously shaken with a vortex, and centrifuged at 500 rpm for 30 sec. CFU were enumerated as described above. Larvae not containing cells due to imperfect application or pathogen clearance were included in the calculation.

To prepare the gut of the insects, the larvae were incubated for 10 min on ice and then placed in PBS buffer cooled to 4˚C for the subsequent intestinal preparation. The larvae were carefully opened making an incision along the coronal plane from the anterior to the posterior end using micro scissors and Dumont forceps (Manufactures D'Outils Dumont SA, Montignez, Switzerland). The entire innards were removed and the digestive tract was dissected, removing all of the fat and organ appendages. For further analysis, the prepared gut was placed in a new tube containing PBS or RNA*later* (Thermo Fisher Scientific Life Technologies GmbH, Darmstadt, Germany).

## Anatomic preparation of the digestive tract of *G. mellonella*

A solution of 2% methylene blue in distilled water was orally applied, and excessive dye was removed by washing. The main body cavity was opened from the head to tail along the ventral midline, and the cuticula was detached and pinned to a styrofoam plate. Under a binocular, the fat body and other organs were separated from the digestive tract. Oral cavity and anus were circumcised and the digestive tract removed.

## Isolation of Y. enterocolitica cells from G. mellonella

Twenty-four hours after infection, the hemolymph of *G. mellonella* was taken with a sterile insulin syringe to a volume of 100 μl, transferred into Eppendorf tubes with 500 μl of resuspension buffer [1 × PBS, 0.5% biotin-free BSA, 10% (v/v) RNAlater (Thermo Fisher Scientific, Langenselbold, Germany)], and stored at 4°C for up to 24 h until further processing. After centrifugation at 9000 × g for 5 min, the pellet was resuspended and incubated in 750 μl of resuspension buffer for 10 min at 4° C under shaking. The suspension contained a *Yersinia*-specific antibody (anti-*Y. enterocolitica* O:9 mouse monoclonal, FITC, PROGEN Biotechnik GmbH, Heidelberg, Germany) that had been diluted 1:250. The sediment obtained by centrifugation at 9,600 × g for 2 min was washed with cold separation buffer (1 × PBS, 0.5% biotin-free BSA, 2 mM EDTA pH 7.4) containing 10% RNAlater, centrifuged, resuspended in the same buffer, and incubated with 10 μl of streptavidin-coupled magnetic beads for 10 min. The washing and centrifugation steps from above were repeated, and the sediment was resuspended in 500 μl of separation buffer containing 10% RNAlater. *Y. enterocolitica* cells were separated using a MACS cell separation system with a LS column (Miltenyi Biotech, Aubum, CA, USA) according to the manufacturer's instructions.

## RNA isolation

RNA was extracted and purified from 1 ml of a *Y. enterocolitica* suspension isolated from an *in vitro* bacterial culture or from 100 μl *G. mellonella* hemolymph by immunomagnetic separation [41] as described above, and stored in RNAlater (Thermo Fisher Scientific). The bacterial pellet was homogenized with PBS, and centrifuged for 5 min at 4°C and 9000 × g. For further cell lysis, 200 μl lysozme (3 mg/ml) were added and vortexed vigorously at 1,300 rpm for 15 min. After the lysate was transferred to a new tube with 50 mg of 0.1 mm Zirconia beads (Biospec, Bartlesville, U.S.A.), 700 μl of RLT buffer were added and the mixture was vortexed three times for 45 sec. After a further centrifugation step, the supernatant was transferred to a new tube, mixed with 470 μl of 100% ethanol, applied to the column, and processed further in the RNeasy Mini Kit according to the manufacturer's instructions (QIAGEN GmbH, Hilden, Germany). RNA quality was assessed using a 2100 Bioanalyser (Agilent, Waldbronn, Germany).

## Transcriptome analysis

Whole-transcriptome RNA library preparation with isolated RNA was performed as described [66]. Briefly, ribosomal RNAs were depleted using the Ribominus Transcriptome isolation Kit (Invitrogen, Darmstadt, Germany), and RNA was fragmented *via* a Covaris sonicator. Following de- and rephosphorylation, the TruSeq Small RNA Sample Kit (Illumina, Munich, Germany) was used for RT-PCR and gel purification of the resulting cDNAs. Libraries were then diluted and sequenced on a MiSeq sequencer (Illumina, Munich, Germany) using a MiSeq Reagent Kit v2 (50 cycles), resulting in 50 bp single-end reads. Illumina FASTQ files were mapped to the reference genome of *Y. enterocolitica* W22703 (accession number PRJEA59689; [14]) using Bowtie for Illumina implemented in Galaxy [67,68]. Artemis was used to visualize

and calculate the number of reads mapping on each gene [69,70]. Gene counts of each library were normalized to the smallest library, and RPKM (reads per kilobase per million mapped reads) values were calculated. Fold changes (FC) between the different conditions were determined.

## Histology

*G. mellonella* larvae were immobilized at 4°C, pinned on cork discs in a stretched position, and placed in 4% neutral buffered formaldehyde at least twelve hours for fixation. Larvae were removed from the cork discs and placed in 3 cm × 2,5 cm × 0,5 cm metal molds filled with 10% liquid agarose. They were oriented by forceps in a straight position with the legs up, and the agarose was allowed to solidify for 30 min at 4°C. After removing excessive agarose, larvae were dissected with a razor blade along the body midline in equal halves using mouth, tail, and legs for orientation. Both parts were placed with the cut surface downwards in cassettes. For paraffin embedding, formaldehyde was removed by rinsing in tap water. Tissues were dehydrated in graded ethanol followed by xylene, infiltrated with paraffin type 1 (Richard Allan Sci., Michigan, USA) in an automatic vacuum infiltration processor (Tissue-Tek VIP 6, Sakura, Staufen, Germany), and embedded in paraffin type 6 (Richard Allan Sci., Michigan, USA). Two µm paraffin sections made by an Epredia HM 355S automatic microtome (Fisher Scientific GmbH, Schwerte, Germany) were collected on charged glass slides and dried at 37°C for 24 to 48 h. Deparaffinised tissue section were stained with hemalaun and eosin for morphological orientation and identification of tissue lesions.

## Immunofluorescence staining

To detect *Y. enterocolitica* by fluorescent-labelled antibodies, successive tissue sections were deparaffinized in xylene and graded ethanols, and antigen-demasked by digestion with 0.1% trypsin for 20 min at 37°C. After washing in PBS, a fluorescent-labelled anti-*Y. enterocolitica* O:9 mouse monoclonal antibody (FITC; PROGEN Biotechnik GmbH, Heidelberg, Germany) was diluted 1:250 in PBS containing 3% BSA and applied for 1 h at room temperature (RT). An anti-*E. coli* polyclonal antibody (FITC) diluted 1:100 in PBS was used in controls (Genetex, Biozol Diagnostica Vertrieb GmbH, Leipziger Straße 4, 85386 Eching, Germany). For TcaA detection, an anti-RFP polyclonal antibody (Invitrogen) was diluted 1:250 in PBS with 3% BSA and incubated for 1 h at RT, followed by application of a goat anti-rabbit IgG (H+L) secondary antibody (Invitrogen) diluted 1:2,000. After a further washing step, tissue sections were covered-slipped with ProLong Diamond Antifade Mountant (Thermo Fisher Scientific, Darmstadt, Germany). Microscopic slides were evaluated on a Zeiss Axio Imager 2 (Carl Zeiss Microscopy GmbH, Göttingen, Germany) using a dual emission filter set, allowing the simultaneous detection of FITC-marked bacteria in the green channel and imaging of the insect tissue through their own fluorescence in the red channel.

## Giemsa staining of hemocytes

*G. mellonella* larvae were picked up and pricked in the aorta of the abdominal area with a sharp cannula. The escaping hemolymph fluid was placed on a microscopic slide. The preparations were dried for about 10 min at RT, followed by fixation for 10 min with methanol or acetone. The sections were placed in dH$_2$O twice for 2 min and stained in the Giemsa solution (5 ml Giemsa stock solution in 80 ml dH$_2$O) for 30 min. After staining, the preparation was differentiated with dH$_2$O (and a few drops of acetic acid) until colouring to mauve, followed by exposure to 96% ethanol to extract colour surplus. Finally, the slides were washed 3 × 2 min in isopropanol, placed 2 × 2 min in xylene, and covered with Eukitt Quick-hardening mounting

medium (Merck KGaA, Darmstadt, Germany). The microscopic slides were evaluated on a Zeiss Axio Imager 2 (Carl Zeiss Microscopy GmbH).

## Supporting information

**S1 Fig.** (A) map of the HE lysis cassette within Tc-PAI$_{Ye}$. (B) Transcriptional response *in vivo* of the genes located on Tc-PAI$_{Ye}$.
(TIF)

**S1 Table. Heat map of differentially regulated *Y. enterocolitica* genes *in vivo*.**
(XLSX)

**S2 Table. Bacterial strains and plasmids used in this study.**
(DOCX)

**S3 Table. Oligonucleotides used in this study.**
(DOCX)

## Acknowledgments

We thank Jana Jagiela und Lisa Wolf for technical assistance, and Marcus Pfau for photographic work.

## Author Contributions

**Conceptualization:** Philipp-Albert Sänger, Elisabeth M. Liebler-Tenorio, Thilo M. Fuchs.

**Data curation:** Philipp-Albert Sänger, Stefanie Wagner, Thilo M. Fuchs.

**Formal analysis:** Philipp-Albert Sänger, Stefanie Wagner, Elisabeth M. Liebler-Tenorio, Thilo M. Fuchs.

**Funding acquisition:** Thilo M. Fuchs.

**Investigation:** Philipp-Albert Sänger, Elisabeth M. Liebler-Tenorio, Thilo M. Fuchs.

**Methodology:** Philipp-Albert Sänger, Elisabeth M. Liebler-Tenorio, Thilo M. Fuchs.

**Project administration:** Thilo M. Fuchs.

**Supervision:** Thilo M. Fuchs.

**Validation:** Elisabeth M. Liebler-Tenorio, Thilo M. Fuchs.

**Visualization:** Elisabeth M. Liebler-Tenorio, Thilo M. Fuchs.

**Writing – original draft:** Philipp-Albert Sänger, Elisabeth M. Liebler-Tenorio, Thilo M. Fuchs.

**Writing – review & editing:** Thilo M. Fuchs.

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
