## [Decision Letter · Decision Letter 0]

10 Oct 2022

Dear Dr Fuchs,

Thank you very much for submitting your manuscript "Dissecting the invasion of Galleria mellonella by Yersinia enterocolitica reveals metabolic adaptations and a role of a phage lysis cassette in insect killing" for consideration at PLOS Pathogens. As with all papers reviewed by the journal, your manuscript was reviewed by members of the editorial board and by several independent reviewers. The reviewers appreciated the attention to an important topic. Based on the reviews, we are likely to accept this manuscript for publication, providing that you modify the manuscript according to the review recommendations.

All of the further requests for this minor revision regard the manuscript text preparation. Please respond to all of the text edits requested by Reviewers #2 and #3 in a revised version. In addition, per request by Reviewer #3, go through the entire manuscript and check grammar usage to improve the overall text. The discussion in particularly needs to be improved for clarity.

Sincerely,

Karla Satchell

Section Editor

PLOS Pathogens

Kasturi Haldar

Editor-in-Chief

PLOS Pathogens

orcid.org/0000-0001-5065-158X

Michael Malim

Editor-in-Chief

PLOS Pathogens

orcid.org/0000-0002-7699-2064

Reviewer Comments (if any, and for reference):

Reviewer's Responses to Questions

**Part I - Summary**

Reviewer #1: (No Response)

Reviewer #2: The author did a great job in reviewing their manuscript. In my opinion, the manuscript is ready for publication after the following minor points have been addressed:

Final sentence of abstract: as a hypothesis, this can of course be left in. But please rephrase to show that this is based on indirect evidence. I would suggest: "..., suggesting that this dual lysis cassette may be an example for a phage-related function that has been adapted for the release of a bacterial toxin". In the current form, this statement is too strong for the data provided in the manuscript.

"Even if this frameshift is not the result of a sequencing error, it obviously does not result in Tc inactivation. As this frameshift was not identified in most other Tc-PAI of yersiniae, we assume our statement to be correct." - Please provide fresh sequencing data for this region of W22703. The region between TcaA and TcaB2 is only 6100 bp, which can easily be covered by a colony PCR and the resulting PCR fragment sequenced. If the TcA component core is indeed split into three subunits, this would represent a novel finding and certainly be interesting for the field.

"This is given below figures 1E-H." - I still cannot relate these to the numbers 1-8 in Figure 1A-1D. Please explain which anatomical features correspond to which numbers.

"The presence of the plasmids in vivo was confirmed by periodic plating on selective and non-selective plates, not revealing differences in cell numbers." - Was a non-transformed control also checked in parallel to confirm selectivity is due to the antibiotic resistance cassette in plasmids rather than the cells acquiring resistance in situ?

"Infections with W22703 delta tccC are not shown to not overload the figure" - Please add the shown panel to the figure, this is important data.

"W22703 delta tccC/pBAD-tccC infections have not been documented by photos." - This is a shame from a consistency viewpoint in relation to the other, photographically documented data. Please change the Figure 3 legend from "are not shown" to "were not documented by photos" for the sake of clarity. In lines 141-152 the W22703 delta tccC/pBAD-tccC is not mentioned in comparison to other tested strains (as per line 133) - can this be fixed?

“This discrepancy suggests that TcaA is involved in adherence to epithelial cells and thus in midgut colonization, without requiring TccC.” - This would be a novel function for Tc toxins - they have so far been considered to function only as holotoxins. Perhaps this novelty should be highlighted.

In Figure 4, the error bars for W22703 delta HE, W22703 delta HE/pACYC-HE, W22703 delta tccC, and W22703 delta tccC/pBAD-tccC are missing, although the legend says the experiments were performed as triplicates. Please address this.

"The FC-value of holin gene is 0.87, thus pointing to a very slight transcription of this lysis gene as discussed, thus preventing cell death". A point well worth bringing up in the discussion, as it points to "a lack of cell lysis as prerequisite for Tc release", as you say in response to point 12.

Reviewer #3: The manuscript is in general improved with greater emphasis on TcA component and per os activity. the authors have attempted to address previous responses in the responses to reviewer sections and only carried through some of these corrections into the revised manuscript.

There are still some areas that if acted on may improve the readability of the revised manuscript , in addition to this a couple of contradictory sentences (as provided below) were noted which need to be clarified. I also think a final comprehensive proofreading of the manuscript is needed to smooth out some sections. Never the less the authors have clearly demonstrated the per os activity and a role of TcA in gut retention. On this note https://www.ncbi.nlm.nih.gov/pmc/articles/PMC1230942/ refer to role of a TcA in gut retention in mouse (might be good to integrate this reference)

They have placed less emphasis on lysis where their transcriptome profile is only partly linked to that of the Tc cluster

**Part II – Major Issues: Key Experiments Required for Acceptance**

Reviewer #1: No further experiments are required.

Reviewer #2: see above

Reviewer #3: (No Response)

**Part III – Minor Issues: Editorial and Data Presentation Modifications**

Reviewer #1: (No Response)

Reviewer #2: see above

Reviewer #3: L78 I still think more information on Lon protease needs to be provided (this might be known to the authors but may not be known to the readers)

L107-118 I would argue if you have done through a serial dose series that seems to be enough data here to determine the Ld50 for a set time period eg but not necessarily 3 days post challenge

L130 perhaps say challenging larvae with LB medium ?

L133 still unsure about pARA leakiness can this be explained better can you place a example from another system I think a paper of Guzman (https://pubmed.ncbi.nlm.nih.gov/7608087/) highlight basal leakyness andcould be referenced

L168 complemented plasmid carried the deleted genes is this correct? I might have though the complement plasmids carried the intact genes as per line 132-133 – please clarify (I might have miss read)

L220 contradicts the statements on Ll167=170 where full complementation is noted ?

L248 might be good to link 12-24 hour in potential role of tcca adhesion also what about holY?

L272 please expand on what the lux operon is so the reader has greater understanding

L275 can we refer to a table etc on the 13 phage genes

The discussion a bit hard to decipher in places, specifically how the examples given link with the findings of the study. Please edit a final time to improve clarity.

L289 majority excreted I might suggest in the absence of TcA the bacteria is unable to be maintain itself within the gut (this might be a better way to word this)

Is l 291 is this correct Pluminescens in the gut – in this respect the nematode typically infects the haemocoel as the first point of entry - please check

L 297 caco 2 gut cells could mention they are human derived

L304 what does readouts? mean please clarify

L343 possibly list the genes that specifically target insects

Table 1 though highlighted in table supp figure 1 I still might suggest highlight relevant TC and lysis genes

Figures are in general well presented

Fig 7 are the scales accurate of are cells observed in Tca minus? Which appear more larger?

PLOS authors have the option to publish the peer review history of their article (what does this mean?). If published, this will include your full peer review and any attached files.

Reviewer #1: No

Reviewer #2: No

Reviewer #3: No

Figure Files:

Data Requirements:

Reproducibility:

References:

---

## [Editor Report · Decision Letter 1]

8 Nov 2022

Dear Dr Fuchs,

We are pleased to inform you that your manuscript 'Dissecting the invasion of Galleria mellonella by Yersinia enterocolitica reveals metabolic adaptations and a role of a phage lysis cassette in insect killing' has been provisionally accepted for publication in PLOS Pathogens.

Best regards,

Karla J.F. Satchell, Ph.D.

Section Editor

PLOS Pathogens

Karla Satchell

Section Editor

PLOS Pathogens

Kasturi Haldar

Editor-in-Chief

PLOS Pathogens

orcid.org/0000-0001-5065-158X

Michael Malim

Editor-in-Chief

PLOS Pathogens

orcid.org/0000-0002-7699-2064
---

## [Editor Report · Acceptance letter]

15 Nov 2022

Dear Dr Fuchs,

We are delighted to inform you that your manuscript, "Dissecting the invasion of Galleria mellonella by Yersinia enterocolitica reveals metabolic adaptations and a role of a phage lysis cassette in insect killing," has been formally accepted for publication in PLOS Pathogens.

Best regards,

Kasturi Haldar

Editor-in-Chief

PLOS Pathogens

orcid.org/0000-0001-5065-158X

Michael Malim

Editor-in-Chief

PLOS Pathogens

orcid.org/0000-0002-7699-2064